# Phases of Rotating Black Objects in $d = 5$ Einstein–Gauss–Bonnet Theory

**Burkhard Kleihaus** [1] , **Jutta Kunz** [1] **and Eugen Radu** [2,*]

1 Institute of Physics, University of Oldenburg, Postfach 2503, D-26111 Oldenburg, Germany
2 Centre for Research and Development in Mathematics and Applications (CIDMA), Campus de Santiago, 3810-183 Aveiro, Portugal
* Correspondence: eugen.radu@ua.pt

**Abstract:** We considered several different classes of asymptotically flat, rotating black objects in $d = 5$ Einstein–Gauss–Bonnet (EGB) theory. These are black holes with two equal-magnitude angular momenta, in which case extremal configurations are studied as well. Numerical evidence is also given for the existence of EGB generalizations of the Myers–Perry black holes with a single plane of rotation and of the Emparan–Reall balanced black rings. All solutions approach asymptotically the Minkowski background and present no singularities outside or on the horizon. The numerical results suggest that, for any mass of the solutions and any topology of the horizon, the rotating configurations exist up to a maximal value of the GB coupling constant, while the solutions with a spherical horizon topology still satisfy the Einstein gravity bound on angular momentum.

**Keywords:** higher-dimensional black holes; black rings; higher curvature theories

## 1. Introduction

In $d = 5$ spacetime dimensions, the Einstein–Gauss–Bonnet (EGB) model provides the most general theory of gravity, which includes higher order curvature terms while keeping the equations of motion to second order [1]. Apart from being of mathematical interest and providing a natural generalization of General Relativity (GR), the Gauss–Bonnet (GB) term appears in the low-energy effective action for the compactification of M-theory on a Calabi–Yau threefold [2,3] and also enters the one-loop corrected effective action of heterotic string theory [4–7].

The Black Hole (BH) solutions of EGB gravity have been studied by various authors, starting with Refs. [8,9], in which a generalization of the Schwarzschild–Tangherlini BH [10] has been found. These solutions possess a variety of new features; for example, their entropy includes a GB contribution [11,12], with the existence of a branch of small static BHs which are thermodynamically stable.

However, the complexity of the EGB theory makes the task of finding solutions beyond those in [8,9] a highly non-trivial problem [13,14]. In particular, no EGB closed form rotating solutions are known yet, and it was proven in [15] that the Kerr–Schild ansatz does not work in this model. Nevertheless, a number of partial results (including perturbative exact solutions [16–18] and numerical non-perturbative results for configurations with symmetry enhancement [19,20]) support the idea that EGB rotating solutions actually exist.

This issue is of special interest, since, as discovered by Emparan and Reall [21], rotation allows in this case for Black Ring (BR) solutions in addition to the generalization of the Kerr BH found by Myers and Perry [22]. This (asymptotically flat, vacuum GR) solution has a horizon with topology $S^2 \times S^1$, while the MP BH has a horizon topology $S^3$. This made clear that a number of well known results in $d = 4$ gravity stop being valid in higher dimensional GR. Therefore, it would be interesting to find out whether the situation persists for other models of gravity.

In this work, we address the question of how the GB term affects the phase structure of several different types of $d = 5$ rotating black objects. We shall first consider BHs with two equal-magnitude angular momenta, extending the results found in Ref. [20] by including the set of extremal solutions. Black objects rotating in a single plane were studied as well, and we report EGB generalizations of both MP BHs and rotating BRs. All solutions were found within a nonperturbative approach, by directly solving the second-order field equations with suitable boundary conditions.

## 2. The Model and the Static Limit

### 2.1. Action, Equations and Scaled Quantities

Working in units with $c = G = 1$, we consider the EGB action in five spacetime dimensions

$$I = \frac{1}{16\pi} \int_{\mathcal{M}} d^5x \sqrt{-g}[R + \alpha L_{\text{GB}}] , \tag{1}$$

where $\alpha$ is the GB coefficient with dimension $(length)^2$. In string theory, the GB coefficient is positive, and this is the only case considered here. $R$ denotes the Ricci scalar, and

$$L_{\text{GB}} = R^2 - 4R_{\mu\nu}R^{\mu\nu} + R_{\mu\nu\rho\sigma}R^{\mu\nu\rho\sigma} \tag{2}$$

is the GB term, with Ricci tensor $R_{\mu\nu}$ and Riemann tensor $R_{\mu\nu\rho\sigma}$.

The variation of the action (1) with respect to the metric tensor yields the EGB equations

$$G_{\mu\nu} + \alpha H_{\mu\nu} = 0 , \tag{3}$$

where

$$
\begin{aligned}
G_{\mu\nu} &= R_{\mu\nu} - \frac{1}{2}g_{\mu\nu}R , \\
H_{\mu\nu} &= 2\left[RR_{\mu\nu} - 2R_{\mu\rho}R_\nu^\rho - 2R_{\mu\rho\nu\sigma}R^{\rho\sigma} + R_{\mu\rho\sigma\lambda}R_\nu^{\rho\sigma\lambda}\right] - \frac{1}{2}g_{\mu\nu}L_{\text{GB}}.
\end{aligned}
$$

The solutions discussed in this work approach asymptotically the $d = 5$ Minkowski spacetime background, with a line element

$$ds^2 = dr^2 + r^2 d\Omega_3^2 - dt^2, \text{ with } d\Omega_3^2 = d\theta^2 + \sin^2\theta d\varphi_1^2 + \cos^2\theta d\varphi_2^2, \tag{4}$$

where $\theta \in [0, \pi/2]$, $(\varphi_1, \varphi_2) \in [0, 2\pi]$, while $r$ and $t$ denote the radial and time coordinate, respectively. Apart from the mass $M$, they possess a nonzero angular momentum $J$ (or two equal angular momenta, $J_1 = J_2 = J$), with $(M, J)$ read as usual from the far field asymptotics of the metric functions $g_{tt}$ and $g_{\varphi_i t}$, respectively. The horizon quantities of main interest are the Hawking temperature $T_H$, event horizon area $A_H$, event horizon velocity $\Omega_H$ (with $\Omega_{H(1)} = \Omega_{H(2)} = \Omega_H$ for BHs rotating in two planes), and also the entropy $S$, which is the sum of one quarter of the event horizon area (the Einstein gravity term) plus a GB correction [12]

$$S = \frac{1}{4} \int_{\Sigma_h} d^3x \sqrt{h}(1 + 2\alpha R_\Sigma), \tag{5}$$

where $h$ is the determinant of the induced metric on the horizon and $R_\Sigma$ is the event horizon curvature. Additionally, the solutions satisfy the first law of thermodynamics,

$$dM = T_H dS + k\Omega_H dJ \tag{6}$$

(with $k = 1$ or $k = 2$ the number of planes of rotation).

In what follows, we shall consider several quantities of interest, normalised with regard to the mass of the solutions and define [1]

$$a_H = \frac{3}{32}\sqrt{\frac{3}{2\pi}}\frac{A_H}{M^{3/2}}, \quad s = \frac{3}{8}\sqrt{\frac{3}{2\pi}}\frac{S}{M^{3/2}}, \quad t_H = 4\sqrt{\frac{2\pi}{3}}T_H\sqrt{M}, \quad j = \frac{3}{4}\sqrt{\frac{3\pi}{2}}\frac{kJ}{M^{3/2}}. \tag{7}$$

The (dimensionless) ratio between the parameter $\alpha$ and the mass is also important, and we define

$$x = c_0\frac{\alpha}{M}, \quad \text{with} \quad c_0 = \frac{3\pi}{4}. \tag{8}$$

*2.2. The Schwarzschild–Tangherlini Solution in EGB Theory*

The static, spherically symmetric EGB BH solution [2] has a relatively simple form [8],

$$ds^2 = \frac{dr^2}{N(r)} + r^2 d\Omega_3^2 - N(r)dt^2, \quad \text{with} \quad N(r) = 1 + \frac{r^2}{4\alpha}\left(1 - \sqrt{1 + \frac{8\alpha(r_h^2 + 2\alpha)}{r^4}}\right). \tag{9}$$

The parameter $r_h > 0$ denotes the event horizon radius, with $N(r) = \frac{2r_h}{r_h^2 + 4\alpha}(r - r_h) + \ldots$ as $r \to r_h$. While $r_H$ can be arbitrarily large, the limit $r_h \to 0$ is nontrivial, with no horizon and $N(r) = 1 - \alpha/(r^2(1 + \sqrt{1 + \frac{\alpha^2}{r^4}}))$ a strictly positive function. However, this configuration is pathological, $r = 0$ corresponding to a naked singularity, with a diverging Ricci scalar.

The expressions of various quantities of interest for the spherically symmetric BH solutions are

$$M = \frac{3\pi}{8}(r_h^2 + 2\alpha), \quad T_H = \frac{r_h}{2\pi(r_h^2 + 4\alpha)}, \quad A_H = 2\pi^2 r_h^3, \quad S = \frac{\pi^2 r_h^3}{2}\left(1 + \frac{12\alpha}{r_h^2}\right). \tag{10}$$

As such, the mass spectrum of these EGB BHs is bounded from below by the mass corresponding to the nakedly singular configuration, $M > 3\pi\alpha/4$, a result which is found to also hold for spinning generalizations.

A straightforward computation leads to the following expression of several scaled quantities, cf. (7):

$$a_H = (1 - x)^{3/2}, \quad s = \sqrt{1 - x}(1 + 5x), \quad t_H = \frac{\sqrt{1 - x}}{1 + x}, \quad \text{with} \quad x = \frac{3\pi\alpha}{4M}, \tag{11}$$

where $0 \leq x \leq 1$. The limit $x = 0$ corresponds to the Schwarzschild–Tangherlini BH in pure Einstein gravity. As $x \to 1$, the minimal mass (nakedly singular) solution is approached, with the (scaled) horizon size, entropy and temperature going to zero. It is interesting to remark that the scaled entropy varies between zero and a maximal value

$$s_{max} = 4\sqrt{2/5} \simeq 2.52892, \tag{12}$$

which is approached for a special configuration with $x = 3/5$ (marked with a black dot in Figure 1, left panel (middle)). Therefore, for given $\alpha$ and a range of $s$, there are two different solutions with the same entropy. At the same time, the scaled horizon area and temperature varies monotonically between one and zero, see the corresponding curves in Figure 1.

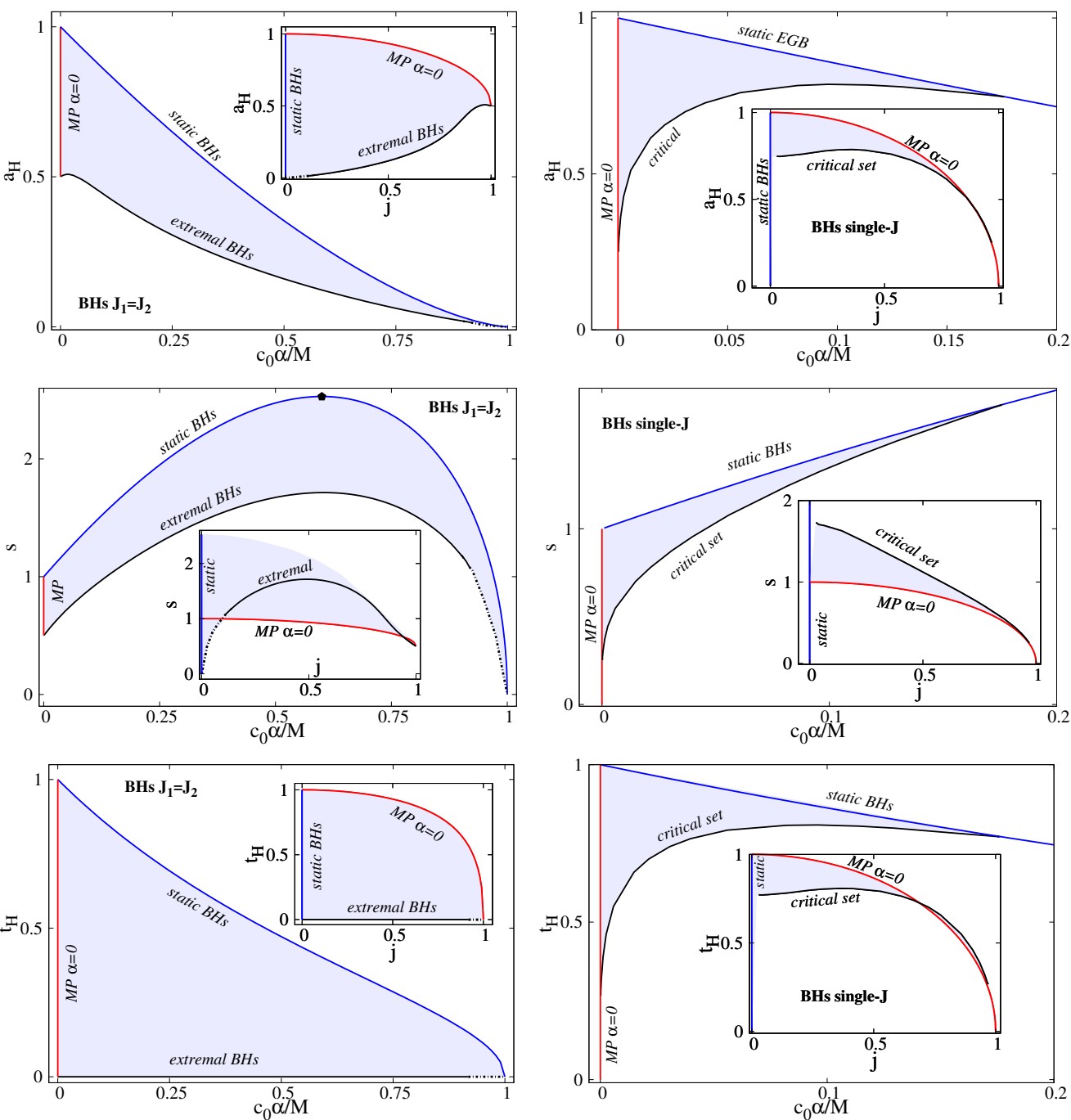

**Figure 1. Left panels:** The domain of existence of the horizon area, entropy and temperature is shown vs. GB parameter *α* and vs. *J* (insets) for EGB black holes with two equal angular momenta. **Right panels:** The investigated region of the parameter space is shown for EGB black holes with a single plane of rotation. All quantities are normalized with regard to the mass of the solutions, while $c_0 = 3\pi/4$.

## 3. Rotating Black Holes: The Case of Equal Angular Momenta

### 3.1. The Ansatz and Particular Cases

For these solutions, the isometry group is enhanced from $\mathbb{R}_t \times U(1)^2$ to $\mathbb{R}_t \times U(2)$ (where $\mathbb{R}_t$ denotes the time translation), a symmetry enhancement which allows us to

factorize the angular dependence of the metric. The line element takes a simple form in terms of the left-invariant one-forms $\sigma_i$ on $S^3$, with

$$ds^2 = f_1(r)dr^2 + \frac{1}{4}f_2(r)(\sigma_1^2 + \sigma_2^2) + \frac{1}{4}f_3(r)(\sigma_3 - 2w(r)dt)^2 - f_0(r)dt^2, \tag{13}$$

where $\sigma_1 = \cos\psi d\bar\theta + \sin\psi\sin\bar\theta d\phi$, $\sigma_2 = -\cos\psi d\bar\theta + \cos\psi\sin\bar\theta d\phi$, $\sigma_3 = d\psi + \cos\bar\theta d\phi$, and we define $2\theta = \bar\theta$, $\varphi_1 - \varphi_2 = \phi$, $\varphi_1 + \varphi_2 = \psi$ (with $\{\theta, \varphi_1, \varphi_2\}$ the angular coordinates in (4)). This geometry describes a fibration of $AdS_2$ over the homogeneously squashed $S^3$ with symmetry group $SO(2,1) \times SU(2) \times U(1)$. The horizon is located at some $r = r_h$ (where $f_0(r_h) = 0$), with the induced horizon metric

$$d\Sigma^2 = h_{ij}dx^i dx^j = \frac{1}{4}\left(f_2(r_h)(\sigma_1^2 + \sigma_2^2) + f_3(r_h)\sigma_3^2\right). \tag{14}$$

For a given mass, the ($\alpha = 0$) MP BHs exhibit a similar behaviour to that found for $d = 4$ Kerr BHs, forming a one parameter family of solutions which interpolates between the static limit ($j = 0$) and an extremal configuration with $j = 1$, $t_H = 0$ and a nonzero horizon area [3]. As expected, all these BHs possess generalizations with $\alpha \neq 0$, a study of the non-extremal solutions being reported in Ref. [20]. Most of the work there has been performed for a metric gauge choice with $f_2(r) = r^2$, the metric functions $f_0(r), f_1(r), f_3(r)$ and $w(r)$ being found numerically as solutions of a complicated set of ordinary differential equations with suitable boundary conditions. A detailed study of these aspects has been reported in Ref. [20] and we shall not repeat it here.

Following the same procedure, we extended the results in Ref. [20], attempting to obtain a complete scan of the domain of existence of the solutions (in particular, for the region with $x > 0.5$, as defined in Equation (8), which was poorly covered in [20]).

Additionally, the results in Ref. [20] strongly suggest that the families of rotating EGB BHs terminate at extremal configurations. Although this special set of solutions was not constructed in [20], the extrapolated results indicated that all relevant quantities remain finite in the extremal limit, while the Hawking temperature vanishes.

This is indeed confirmed by the results below, which are found by extending the methods in [20], and constructing directly the extremal BHs in EGB theory. These solutions are found for a form of the metric ansatz (13), with $f_1(r) = e^{2F_1(r)}/B_1(r)$, $f_2(r) = e^{2F_2(r)}u(r)$, $f_3(r) = e^{2(F_2(r)+F_3(r))}u(r)B_3(r)$, $f_0(r) = e^{2F_1(r)}B_2(r)$, $w(r) = w_0(r) + W(r)$, a parametrization which contains four unknown functions $F_1(r)$, $F_2(r)$, $F_3(r)$ and $W$, as well as the background functions

$$B_1(r) = \frac{r^2}{u(r)}, \quad B_2(r) = \frac{r^4}{u(r)^2 + a^4}, \quad B_3(r) = 1 + \frac{a^4}{u(r)^2}, \quad w_0(r) = \frac{\sqrt{2}a^3}{u(r)^2 + a^4},$$

with $u(r) = r^2 + a^2$ and $a > 0$ an input parameter [4].

In this approach, the extremal horizon is located at $r = 0$, where one can construct an approximate form of the solutions as a power series in $r$. A similar approximate solution can be found for large $r$ (with, e.g., $F_1 = c_t/r^2 + \dots$ and $W = c_w/r^4 + \dots$), which reveals the existence of two free constants $c_t$ and $c_w$. The solutions that smoothly interpolate between these asymptotics are found by using similar methods to those described in [20], and by solving numerically the equations for $(F_i, W)$ with suitable boundary conditions. The quantities of interest are computed from the numerical output, with

$$M = \frac{3\pi}{4}(a^2 - c_t), \quad J = \frac{\pi}{4}(c_w + \sqrt{2}a^3), \quad A_H = 4\pi^2\frac{a^3}{\sqrt{2}}e^{3F_2(0)+F_3(0)}, \tag{15}$$

$$S = \pi^2\frac{a^3}{\sqrt{2}}e^{3F_2(0)+F_3(0)} + 4\pi^2\sqrt{2}\alpha a e^{F_2(0)+F_3(0)}\left(2 - e^{2F_3(0)}\right).$$

### 3.2. The Domain of Existence and Attractors

In Figure 1 (left panels), we plot the domain of existence of solutions (shaded blue region), as resulting from the extrapolation of around one thousand data points into the continuum. The figure shows that this region is delimited by [5]: (*i*) the set of static BHs discussed in Section 2.2 (blue curve); (*ii*) the set of extremal BHs (black curve); and (*iii*) the set of $\alpha = 0$ GR solutions corresponding to the MP BHs (red curve). As one can see, the inequality $j \leq 1$ (which is satisfied by the $\alpha = 0$ BHs) still holds in the EGB theory. Moreover, the upper bound found for static BHs $\alpha < 4M/(3\pi)$ is still valid for spinning solutions.

Figure 1 (left) also includes the sets of extremal solutions discussed above (which also emerge as limits of the configurations in Ref. [20]). As $\alpha$ is varied for a given mass, these configurations connect the extremal MP limit with a critical configuration. The limit is difficult to approach, since the integration of the equations is becoming increasingly difficult. Nevertheless, we conjecture that this limit of the extremal set of solutions corresponds to the static singular solution discussed in Section 2, with $M = \frac{3\pi}{4}\alpha$ and $J = A_H = S = 0$. As such, the corresponding curves for extremal solutions in Figure 1 (left) have been extrapolated to this point (dotted black line).

Apart from the numerical results, another indication supporting this conjecture (together with several analytical results) comes from the study of an exact EGB solution describing a rotating squashed $AdS_2 \times S^3$ spacetime, which corresponds to the neighbourhood of the event horizon of an extremal BH. The corresponding metric Ansatz is given again by (13), with $f_0 = v_1 r^2$, $f_1 = v_1/r^2$, $f_2 = v_2$, $f_3 = v_2 v_3$ and $w = -kr$, the constant parameters $v_1, v_2, v_3$ and $k$ being found by solving the EGB equations. This results in a single parameter family of solutions [20], which takes a relatively simple form in terms of $v_3$ (which measures the relative squashing of the $S^3$-sector in (13), with $0 \leq v_3 \leq 2$):

$$v_1 = \frac{(v_2 - 4\alpha(3v_3 - 4))(3v_2 + 4\alpha(4 - v_3))}{2(4 - v3)(3v_2 + 4\alpha(8 - 6v_3))}, \quad k = (4 - v_3)\sqrt{v_2 v_3}\frac{\pi v_1}{2J}, \tag{16}$$

with

$$v_2 = \frac{4\alpha}{v_3 - 2}\left(2v_3^2 - 7v_3 + 4 - \sqrt{5v_3^4 - 34v_3^3 + 73v_3^2 - 56v_3 + 16}\right). \tag{17}$$

The attractor formalism allows us to compute the expressions for the angular momentum, event horizon area and entropy of the solutions, with

$$J = \frac{\pi}{4}v_2 v_3\sqrt{(4 - v_3)(v_2 + 4\alpha(4 - 3v_3))}, \tag{18}$$

$$A_H = 2\pi^2 v_2\sqrt{v_2 v_3}, \quad S_{(extremal)} = \frac{\pi^2}{2}\sqrt{v_2 v_3}(v_2 + 4\alpha(4 - v_3)). \tag{19}$$

Therefore, the special configuration with $v_3 = 0$ corresponds to the critical limiting solution, which has $v_1 = \alpha$, $v_2 = 0$ and $A_H = S = J = 0$.

The connection of the above results with the extremal BH solutions is straightforward, via the following identification,

$$v_1 = \frac{1}{4}e^{2F_1(0)}a^2, \quad v_2 = e^{2F_2(0)}a^2, \quad v_3 = 2e^{2F_3(0)}, \tag{20}$$

and is used (together with (16)–(18)) to check the accuracy of the numerical results.

## 4. Black Objects Rotating in a Single Plane: Holes and Rings

### 4.1. The Ansatz and Quantities of Interest

The case of BHs with two equal-magnitude angular momenta is rather special, since generically $J_1 \neq J_2$. However, in the absence of the symmetry enhancement, this results in a set of highly nonlinear coupled partial differential equations, which are difficult to study. In what follows, we shall simplify the problem, restricting to configurations with a single plane of rotation. Two different classes of solutions are considered in this case, corresponding to EGB generalizations of (singly spinning) MP BHs (with an $S^3$ event horizon topology) and of Emparan–Reall BRs (with an $S^2 \times S^1$ event horizon topology).

Both types of configurations are constructed within a metric ansatz [6] with five unknown functions $(f_i, w)$:

$$ds^2 = f_1(r,\theta)(dr^2 + r^2 d\theta^2) + f_2(r,\theta)(d\varphi_1 - w(r,\theta)dt)^2 + f_3(r,\theta)d\varphi_2^2 - f_0(r,\theta)dt^2. \tag{21}$$

For both BHs and BRs, the event horizon is localized at constant radius $r = r_h$, where $f_0(r_h) = 0$. Expanding the EGB equations in the vicinity of the horizon in powers of $r - r_h$, one finds $f_i(r,\theta) = f_{i0}(\theta) + f_{i2}(\theta)(r - r_h)^2 + O(r - r_h)^3$, $w(r,\theta) = \Omega_H + w_2(\theta)(r - r_h)^2 + O(r - r_h)^3$ (where the functions $f_{ik}(\theta), w_2(\theta)$ are solutions of a set of nonlinear second order ordinary differential equations and $f_{00}(\theta) = 0$), which leads to an event horizon metric

$$d\Sigma^2 = h_{ij}dx^i dx^j = f_{10}(\theta)r_h^2 d\theta^2 + f_{20}(\theta)d\varphi_1^2 + f_{30}(\theta)d\varphi_2^2. \tag{22}$$

For any horizon topology, the Hawking temperature, horizon area and the entropy of an EGB solution read

$$T_H = \frac{1}{2\pi}\sqrt{\frac{f_{02}}{f_{10}}}, \ A_H = 4\pi^2 r_h \int_0^{\pi/2} d\theta \sqrt{f_{10}f_{20}f_{30}}, \ S = \pi^2 r_h \int_0^{\pi/2} d\theta \sqrt{f_{10}f_{20}f_{30}}(1 + 2\alpha R_\Sigma),$$

with

$$R_\Sigma = \frac{1}{2r_h^2 f_{10}}\left(\left(\frac{f_{20,\theta}}{f_{20}} + \frac{f_{30,\theta}}{f_{30}}\right)\frac{f_{10,\theta}}{f_{10}} + \frac{f_{20,\theta}^2}{f_{20}^2} + \frac{f_{30,\theta}^2}{f_{30}^2} - \frac{f_{20,\theta}f_{30,\theta}}{f_{20}f_{30}} - \frac{2f_{20,\theta\theta}}{f_{20}} - \frac{2f_{30,\theta\theta}}{f_{30}}\right).$$

BHs and BRs are distinguished by the boundary conditions they satisfy at $\theta = 0$. For BRs, one imposes $f_3 = \partial_\theta f_0 = \partial_\theta f_1 = \partial_\theta f_2 = \partial_\theta w = 0$ for $r_h \leq r < r_b$, and $f_2 = \partial_\theta f_0 = \partial_\theta f_1 = \partial_\theta f_3 = \partial_\theta w = 0$ for $r \geq r_b$ (with $r_b > r_h$ an input parameter roughly corresponding to the ring's $S^1$ radius [23,24]). The generalizations of the MP BHs have $f_2 = \partial_\theta f_0 = \partial_\theta f_1 = \partial_\theta f_3 = \partial_\theta w = 0$ for any $r \geq r_h$. For both types of solutions, the conditions satisfied by the metric functions at $\theta = \pi/2$ are $f_3 = \partial_\theta f_0 = \partial_\theta f_1 = \partial_\theta f_2 = \partial_\theta w = 0$.

As $r \to \infty$, the Minkowski spacetime background (4) is recovered, with $f_0 = f_1 = 1$, $f_2 = r^2 \sin^2\theta$, $f_3 = r^2 \cos^2\theta$, $w = 0$. The mass $M$ and the angular momentum $J$ of solutions are read from the asymptotic expansion of the metric functions, $f_0 = 1 - 8M/3\pi r^2 + \dots$, $w = 4J/\pi r^4 + \dots$.

A crucial ingredient of our approach is to use a set of background functions which automatically take into account the sets of boundary conditions on the boundaries that determine the topology of the horizon. One defines $f_i = F_i f_i^{(b)}$ and $w = F_4 + w^{(b)}$, where $f_i^{(b)}$ and $w^{(b)}$ are the functions of the corresponding solution in Einstein gravity [7]. It follows that the boundary conditions satisfied by $F_i$ are $\partial_r F_i = 0$ at the horizon, $F_i = 1$ ($i = 0, \dots, 3$), $F_4 = 0$ at infinity, and $\partial_\theta F_i = 0$ on the symmetry axes ($\theta = 0, \pi/2$). We then employ a numerical scheme developed in [23,24] which uses a Newton–Raphson method to solve for the $F_i$, whilst ensuring that all the EGB equations are satisfied. Mapping spatial infinity to the finite value $\bar{r} = 1$ via $\bar{r} = 1 - r_h/r$, the numerical errors for the functions are estimated to be in the order of $10^{-3}$. The reader is referred to Appendix B of [23] for details of the procedure.

*4.2. The Solutions*

Detailed discussions of the properties of the MP BH and BR solutions in Einstein gravity have appeared in various places in the literature, see e.g. the review work [26]. Here, we shall briefly mention only some features which occur later when discussing the numerical EGB generalizations. For a given mass, the MP BHs describe a one-parameter family of solutions which interpolate between the static BHs and a maximally rotating configuration which is singular, with $j = 1$ and zero temperature and horizon area [8]. The picture for BRs is more complicated, with the existence of two branches of solutions which branch off from a cusp at $j = j_{(min)} \simeq 0.918$, $a_H = a_{H(max)} \simeq 0.354$ and $t_H \simeq 0.707$. One of these branches corresponds to thick BRs and has a small extent, meeting at $(j, aH) = (1, 0)$ the singular MP solution. No upper bound on $j$ exists for the thin BRs' branch, which at large angular momentum effectively becomes boosted black strings. Moreover, in the region $j_{(min)} < j < 1$, three black objects with the same mass and angular momentum coexist, thus violating BH uniqueness.

Starting from the respective solutions in Einstein gravity ($F_0 = F_1 = F_2 = F_3 = 1$, $F_4 = 0$), we have generated branches of BHs and BRs by increasing the GB coupling constant $\alpha$ from zero, while keeping the parameters $r_h, w_h$ (and $r_b$ for BRs) fixed. To assure that the solutions are regular, we have monitored a number of invariant quantities such as the Ricci and Kretschmann scalars. All solutions we have found are finite in the full domain of integration, in particular at $r = r_h$ and at $\theta = 0, \pi/2$. Additionally, let us mention that, as with the Einstein gravity case, the generic BRs describe *unbalanced* configurations (which thus would possess one extra parameter). As such, for given $(r_h, r_b)$, solutions without conical singularities are found for a single value of $w_h$ only. All BRs reported in this work are balanced BRs.

For the case of BHs with a spherical event horizon topology, the results of the numerical investigation are shown in Figure 1 (right panel) as resulting from several hundreds of data points. Let us mention that, different from the previous Section, this covers only partially the full domain of existence of the solutions. In particular, we could not construct accurate enough rotating solutions emerging from static solutions close to the singular one with $x = 1$, which we think is only a numerical issue. We also remark that all generalizations of the MP BHs we have investigated satisfy the $j < 1$ bound, which is likely to hold in the presence of a GB term.

As with the BHs with two equal angular momenta, two boundaries of the domain displayed in Figure 1 (right panels) are provided by: ($i$) the (Einstein gravity) MP solutions; and ($ii$) the static (spherically symmetric) BHs discussed in Section 2. In addition, there is also ($iii$) the set of critical solutions, which is approached for a maximal value of the GB coupling constant. Unfortunately, due to severe numerical difficulties, we could not clarify the meaning of this critical set. There the numerics fail to converge, without an obvious pathological behaviour of the solutions in their vicinity (see also the comments at the end of this Section). However, it is tempting to conjecture that this set emerges at the critical (singular) MP solution, and ends in the static singular solution in the EGB model (two regions which were not possible to investigate numerically).

We have also managed to construct EGB generalizations of the Emparan–Real (balanced) BRs, several results being shown in Figure 2. The numerical investigation was less systematic in this case, and we did not aim to scan their domain of existence. We only remark that, as the GB term is added, the two branches of BRs mentioned above persist for small values of $\alpha$ (together with the corresponding BHs with an $S^3$ topology of the horizon). Thus, non-uniqueness *persists* in EGB theory. We also notice [9] the existence of BR solutions violating the GR bounds, i.e., with $j < j_{(min)}$ and $a_H > a_{H(max)}$, see Figure 2. However, rather unexpectedly, in our calculations, both the BH branch and the thick BR branch terminate before an extremal singular configuration with a vanishing area is reached, and also the thin BR branch cannot be extended to (arbitrarily) large values of $j$, see Figure 2 (right panel).

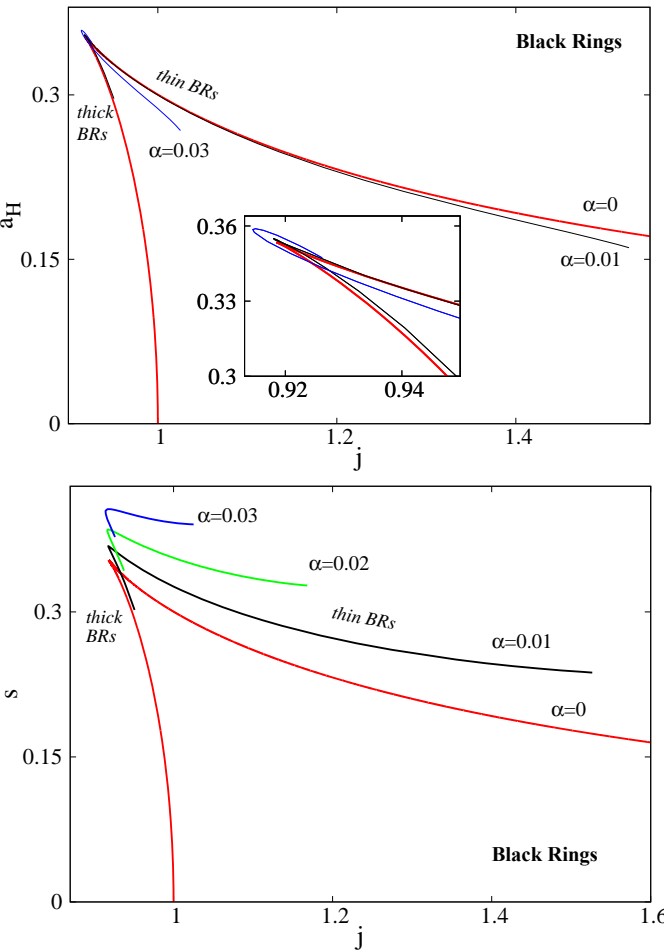

**Figure 2.** The reduced horizon area $a_H$ and entropy $s$ are shown as a function of the reduced angular momentum $j$ for balanced black ring solutions in Einstein–Gauss–Bonnet theory with several values of the GB parameter $\alpha$.

We therefore conjecture that, as $\alpha$ increases, the domain of existence of all three branches of black objects decreases. Consequently, the region where BHs and BRs coexist also decreases with $\alpha$. As such, beyond a first critical value of $\alpha$, BHs and BRs no longer coexist, while beyond a second critical value only BHs persist.

The conclusion that (balanced) rotating BRs exist only up to a maximal value of the GB coupling should not come as a surprise, though, since such behaviour was already found for static BRs in EGB theory [23]. There, the existence of a maximal $\alpha$ follows from conditions on the metric functions for a regular horizon [23], being analogous to that found in the black string case [29]. For balanced thin BRs, our numerical results indicate that, indeed, a similar condition should hold and thus impose a maximal value for $\alpha$. For BHs and balanced thick BRs, however, a different condition should impose a maximal value of $\alpha$ and limit their domain of existence. While we have not been able to clarify its origin, and simply noticed its presence, we conjecture that this could be explained by investigating the expressions of higher order terms in the near-horizon expansion of the solutions [10].

## 5. Further Remarks

The main purpose of this paper was to present a preliminary discussion of three different classes of rotating Black Holes (BHs) in $d = 5$ EGB theory. These are the generalizations of the Myers–Perry (MP) BHs with one and two (equal-magnitude) angular momenta, and of the Emparan–Reall balanced Black Rings (BRs).

The results here strongly suggest that, as expected, any Einstein gravity solution possesses generalizations with a GB term. Additionally, the upper bound (for a given mass)

on the value of the GB coupling constant $\alpha$ found in the static case holds as well for rotating solutions. Moreover, the solutions with a spherical horizon topology still satisfy the GR bound on the angular momentum, $j \leq 1$.

For the case of doubly spinning BHs, the inclusion of a GB term in the action does not affect most of the qualitative features of the known MP solutions. This holds as well for the extremal EGB BHs, which are reported here for the first time in the literature. Although our results for the singly spinning black objects are only partial, they indicate the existence of a different type of critical behaviour of the solutions at maximal $\alpha$, which we could not yet clarify. Nevertheless, we have found that the non-uniqueness of solutions (with the existence of three different objects with the same mass and angular momentum) holds in EGB theory for small enough $\alpha$ only. Further progress in the study of EGB solutions with a single $J$ seems to require a different numerical scheme.

Finally, it would be interesting to compare the results in this work with those found in [18] within a perturbative approach.

**Author Contributions:** B.K., J.K. and E.R. contributed equally to this work. All authors have read and agreed to the published version of the manuscript

**Funding:** The work of B.K. and J.K. was funded by DFG Research Training Group 1620 *Models of Gravity* and DFG project Ku612/18-1. The work of E.R. was funded by the Alexander von Humboldt Foundation, and also by the projects CERN/FIS-PAR/0027/2019, PTDC/FIS-AST/3041/2020, CERN/FIS-PAR/0024/2021 and 2022.04560.PTDC. His work was also supported by the Center for Research and Development in Mathematics and Applications (CIDMA) through the Portuguese Foundation for Science and Technology (FCT—Fundação para a Ciência e a Tecnologia), references UIDB/04106/2020 and UIDP/04106/2020. This work has further been supported by the European Union's Horizon 2020 research and innovation (RISE) programme H2020-MSCA-RISE-2017 Grant No. FunFiCO-777740 and by the European Horizon Europe staff exchange (SE) programme HORIZON-MSCA-2021-SE-01 Grant No. NewFunFiCO-101086251.

**Data Availability Statement:** This manuscript has no associated data or the data will not be deposited. Data will be available from the authors upon request.

**Conflicts of Interest:** The authors declare no conflict of interest.

## Notes

1　Various numerical factors in Equation (7) have been chosen such that $t_H = a_H = s = 1$ in the static limit with $\alpha = 0$, while the maximal value for Einstein gravity BH solutions is $j = 1$.

2　Static EGB solutions with a $S^2 \times S^1$ horizon topology (i.e., BRs) are also known to exist [23], although not in closed form. However, these solutions (still) possess a conical singularity, and thus are physically less interesting.

3　The $\alpha = 0$ MP solution can be written in the form (13), with the expression of the functions $f_i(r)$ and $w(r)$ given, e.g., in Section 2.3 of Ref. [20]. Additionally, this solution has $a_H = s = \frac{1}{2}\left(1 + \sqrt{1-j^2}\right)$, and $t_H = 2\sqrt{1-j^2}/(1 + \sqrt{1-j^2})$.

4　The limit $F_0 = F_1 = F_2 = W = 0$ corresponds to the extremal MP solution in Einstein gravity.

5　Note, that a part of the boundary of the $(j, s)$-domain consists of a configurations with maximal entropy, which do not coincide with the other sets of limiting solutions.

6　Note that the line element (21) can be employed as well in the study of solitonic compact objects, in which case the range of the radial coordinate is $0 \leq r < \infty$. Such configurations possess no horizon ($f_0(r, \theta) \neq 0$) and satisfy a specific set of boundary conditions at the origin, $r = 0$ (with $f_2 = f_3 = W = 0$ and $\partial_r f_1 = \partial_r f_0 = 0$), while the boundary conditions at $\theta = 0, \pi/2$ and at $r \to \infty$ are similar to those employed for BHs with spherical horizon topology. Additionally, one remarks that the static limit of the line-element (21) results in the Schwarzschild–Tangherlini-EGB solution in isotropic coordinates, i.e., with a different radial coordinates than in (9).

7　Both the MP BH and the Emparan–Reall balanced BR can be written in the coordinate system (21), with a complicated expression of the metric functions [25].

8　The MP solution with rotation in a single plane has $a_H = t_H = \sqrt{1-j^2}$. For the corresponding BRs one finds instead $j = (1 + x^2)^3/(4x(1 + x^4))$, $a_H = x(x^2 - 1)/(1 + x^4)$ and $t_H = (x^2 - 1)/(2x)$, with $x \geq 1$.

9　One remarks that the horizon area of BRs, when considered as a function of angular momentum (at fixed mass), exhibits a "loop" in the vicinity of $j_{min}$ (instead of a spike, as for $\alpha = 0$), see the inset in Figure 2. The existence of such loops in the phase diagram of spinning solutions has also been noticed in some $d = 4$ models with non-Abelian matter fields [27,28].

[10] We mention that the four dimensional BHs in EGB-dilaton theory [30,31] also possess a set of critical solutions where the numerics stop converging. However, in that case, it was possible to explain this feature with a study of the second order terms in the near-horizon expansion of the solutions.

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
