# Peer review of "Phases of Rotating Black Objects in d = 5 Einstein–Gauss–Bonnet Theory"

_universe, doi:10.3390/universe9040156_

Round 1

Reviewer 1 Report

The manuscript 'Phases of rotating black objects in d=5 EGB theory' of B. Kleihaus, J. Kunz,  E. Radu deals with studying of rotating BHs with different topology. The BH metrics anzantz can be used during consideration of compact object of different nature. And in this relation I suppose that it is necessary to  write additional criteria for distinguishing black holes from compact objects with the same metric. 

The manuscript can be published after the consideration the criteria for distinguishing BH from compact object with the same metric. 

Author Response

We would like to thank the Referee for the careful reading of our work.

In the revised version of the manuscript we have added a remark   on the additional criteria for distinguishing black holes from compact objects with the same metric
(footnote 6 on page 8 of the manuscript).

Reviewer 2 Report

Please find the report in the attachment.

Author Response

We would like to thank the Referee for 
the careful reading of our work and the constructive remarks.

Following the report, we have  added the  
reference papers for the Schwarzschild-Tangherlini solution in Eq.(2.9) in Section 2.2.
Also, we have marked  with a black dot in Fig.1 (left panel, middle)   the special configuration with $x=3/5$ 
and added a comment on this, after the Eq. (2.12).

Reviewer 3 Report

The paper considers the Gauss-Bonnet modification of general relativity and explains several different classes of asymptotically flat, rotating black hole objects in x Einstein Gauss-Bonnet theory. These are the first black holes with two equal-magnitude angular momenta, in which case extremal configurations are studied as well. Numerical evidence is also given for the existence of Einstein Gauss-Bonnet generalizations of the Myers-Perry black holes with a single plane of rotation and of the Emparan-Reall balanced black rings.

The paper is well organized and the calculations are seemed to be legitimate. It has many important cornerstone physical insights. However, there are a few suggestions of mine, which should be considered before any further statement.

Comments and Suggestions for Authors

1.    The main content is not properly discussed and many of the references are omitted or missing when presenting well-known results on the topic. The authors do not describe a clear motivation for conducting this work in the introduction section.

2.    The introduction attempts to motivate the study of Einstein's Gauss-Bonnet gravity, which is not sufficient. This can be extended to further descriptions of general modified gravity theories.

3.    Furthermore, the authors should compare their results with some observed stellar structures.  There is not so much presented in terms of the new physical implications of the considered modified gravity theory in the case of black holes or other compact objects. This point should be mentioned and discussed.

4.    Also regarding f(G) gravity and the Einstein Gauss-Bonnet gravity, the authors could also mention several applications of these two theories for example f(G) gravity see http://doi.org/10.1103/PhysRevD.92.041302, http://doi.org/10.1103/PhysRevD.92.124027, http://doi.org/10.1093/mnras/stab2062 ,  http://doi.org/10.1016/j.dark.2022.101015 and for Einstein Gauss Bonnet string motivated the authors could mention regarding black holes and astrophysical solutions http://doi.org/10.1103/PhysRevLett.107.271101, http://doi.org/10.1103/PhysRevD.97.084037, http://doi.org/10.1002/andp.202200252.  

After these changes, the article can be accepted for publication.

Author Response

We would like to thank the Referee for reading our work and the comments.